# Evolved histone tail regulates 53BP1 recruitment at damaged chromatin

Jessica L. Kelliher[1,2], Melissa L. Folkerts ●[3,4], Kaiyuan V. Shen[3,4], Wan Song[5], Kyle Tengler[5], Clara M. Stiefel[5], Seong-Ok Lee[1], Eloise Dray ●[6], Weixing Zhao ●[6], Brian Koss[2], Nicholas R. Pannunzio ●[3] ✉ & Justin W. Leung ●[1,5] ✉

The master DNA damage repair histone protein, H2AX, is essential for orchestrating the recruitment of downstream mediator and effector proteins at damaged chromatin. The phosphorylation of H2AX at S139, γH2AX, is well-studied for its DNA repair function. However, the extended C-terminal tail is not characterized. Here, we define the minimal motif on H2AX for the canonical function in activating the MDC1-RNF8-RNF168 phosphorylation-ubiquitination pathway that is important for recruiting repair proteins, such as 53BP1 and BRCA1. Interestingly, H2AX recruits 53BP1 independently from the MDC1-RNF8-RNF168 pathway through its evolved C-terminal linker region with S139 phosphorylation. Mechanistically, 53BP1 recruitment to damaged chromatin is mediated by the interaction between the H2AX C-terminal tail and the 53BP1 Oligomerization-Tudor domains. Moreover, γH2AX-linker mediated 53BP1 recruitment leads to camptothecin resistance in H2AX knockout cells. Overall, our study uncovers an evolved mechanism within the H2AX C-terminal tail for regulating DNA repair proteins at damaged chromatin.

DNA damage occurs in the context of chromatin. Chromatin modifications in response to DNA damage are crucial to choreograph DNA damage response (DDR) pathways[1]. In particular, the roles of the DDR histone, H2AX, have been extensively studied in the past two decades[2–15]. Upon DNA damage, PIKK3-mediated H2AX phosphorylation at serine 139 (γH2AX) initiates a cascade of signaling events to ensure timely and accurate DNA damage repair[5,7,16–18]. The γH2AX signal spreads 1-2 megabase pairs from the site of DNA breaks[6]. This amplified signal facilitates the recruitment of mediator proteins, such as Mediator of DNA Damage Checkpoint 1 (MDC1)[19,20] and microcephalin 1 (MCPH1)[21] and effector proteins to the damage sites[18]. In humans, the H2AX-mediated DNA damage response (DDR) pathway is composed of histone H2AX, MDC1, and the ubiquitin E3 ligases RNF8 and RNF168[19,20,22–27], which together generate an H2A-ubiquitination recognition module for downstream DNA repair protein accumulation, including 53BP1 and the BARD1-BRCA1 complex[26–28].

Interestingly, human H2AX amino- and carboxyl-terminal tails are not highly evolutionarily conserved. The N-terminal tail lysine residues have evolved into an RNF168 substrate[29], which is a key damaged chromatin docking platform for repair proteins, including 53BP1 and BARD1[30,31]. Although the C-terminal SQ[E/D]Φ motif is evolutionarily conserved, the H2AX C-terminal linker region is diverse and has never been systematically characterized. The human H2AX C-terminal linker region has 12 and 16 more amino acids compared to yeast and giardia orthologs, respectively[3]. Additionally, while the γH2AX phosphorylation SQ[E/D]Φ motif is a signature of DNA damage and is indispensable for DNA repair signaling, the minimal molecular requirements of H2AX for the DDR pathway is still unclear.

[1]Department of Radiation Oncology, University of Arkansas for Medical Sciences, Little Rock, AR 72205, USA. [2]Department of Biochemistry and Molecular Biology, University of Arkansas for Medical Sciences, Little Rock, AR 72205, USA. [3]Department of Medicine, Division of Hematology/Oncology, University of California, Irvine, Irvine, CA 92697, USA. [4]Department of Biological Chemistry, University of California, Irvine, Irvine, CA 92626, USA. [5]Department of Radiation Oncology, University of Texas Health and Science Center, San Antonio, TX 78229, USA. [6]Department of Biochemistry and Structural Biology, University of Texas Health and Science Center, San Antonio, TX 78229, USA. ✉e-mail: nrpann@hs.uci.edu; Leungj@uthscsa.edu

Here, we define the minimal motif at the H2AX C-terminus that is required for activation of the DDR phosphorylation-ubiquitination cascade. We also uncover a molecular regulation for 53BP1 ionizing radiation-induced foci (IRIF) formation at an epigenetic level that is independent of the canonical MDC1-RNF8-RNF168 axis. Mechanistically, we identify the 53BP1 domains responsible for its DNA damage recruitment via interaction with the H2AX linker region. Our study provides insights into the molecular regulation of 53BP1 function at damaged chromatin and sheds light on the evolution of histone H2AX and 53BP1 regulation with the DDR.

## Results

### The H2AX-(X)pSQEY is the minimal DDR activating motif
Canonical H2A and H2AX share 95% homology with the latter having a distinctive C-terminal tail (Fig. 1a)[14] that is phosphorylated on serine 139 upon DNA damage to yield γH2AX. Despite the fact that γH2AX was discovered more than two decades ago[5,6], the minimal sequence that is required for the DDR is not defined. Moreover, the functional roles of the evolved extended C-terminal tail are not characterized beyond the pSQEY motif (Fig. 1b).

To systematically define the minimal molecular requirement within the H2AX C-terminal tail for DDR signaling, we generated serial deletion mutants (Supplementary Fig. 1a) within the H2AX C-terminal linker region between the histone fold region and serine 139. Western blot analysis showed that deletion of amino acids 120–136 (Δ120–136) has comparable γH2AX signal with wildtype while deletion of 120–137 (Δ120–137) has no detectable γH2AX signal using monoclonal antibody (JWB301, EMD Millipore, 05-636) (Fig. 1d) suggesting that Δ120-137 either lost the antibody epitope or reduced S139 phosphorylation.

We then used H2AX KO cells with a complementation system and ionizing radiation-induced foci (IRIF) of γH2AX, MDC1, and 53BP1 as biological readouts (Fig. 1c) to investigate the minimal H2AX C-terminal tail requirements for repair protein assembly at DNA damage sites. In brief, we re-express H2AX (wildtype or mutant) with emGFP-expression vectors in U2OS H2AX KO cells. Since both MDC1 and 53BP1 do not form foci in H2AX KO cells[32], we can use the untransfected cells as negative control while wildtype reconstitution served as a positive control. As expected, the control groups with wildtype, S139A and empty vector reconstitutions showed positive and negative immunofluorescence IRIF staining, respectively, for all three markers (Fig. 1e, f and Supplementary Fig. 1b). H2AX Δ120-136 showed no difference in γH2AX or MDC1 and 53BP1 IRIF formation. Consistent with the western blot data, H2AX Δ120-137 reconstitution showed no γH2AX immunofluorescence signal (Fig. 1e), yet there were no discernable defects in MDC1 or 53BP1 IRIF formation compared to wildtype, suggesting that Q137 is an essential epitope for the monoclonal γH2AX antibody and this deletion mutant retains the ability to assemble repair proteins at damaged chromatin. H2AX Δ120-138 failed to rescue MDC1 and 53BP1 IRIF formation in H2AX KO cells implicating that the SQEY motif alone is not sufficient to recruit MDC1. While we found the H2AX C-terminal ASQEY sequence is sufficient for MDC1 and 53BP1 recruitment in H2AX Δ120-137, it remained unclear if alanine 138 confers specificity to their IRIF formation or if it only represents a minimal tail length requirement. To test this, we generated A138T or A138L mutants on the Δ120-137 backbone. We found that these two mutants were able to rescue MDC1 and 53BP1 foci, demonstrating that there are no sequence specificity requirements at the amino-terminus of S139 for MDC1 recruitment to DNA damage sites (Fig. 1e, f and Supplementary Fig. 1c).

To prove that (X)pSQEY is the minimal motif for the DDR pathway, we generated chimera proteins by fusing the SQEY sequence to the histone H2A variants, H2AZ and macroH2A1 (Fig. 2a). These chimera proteins, but not the wildtype counterparts, were able to fully rescue MDC1 and 53BP1 IRIF in H2AX KO cells (Fig. 2b, c, and Supplementary

Fig. 1d–f). Together, these data showed that the H2AX-(X)pSQEY motif is the minimum sequence for the phosphorylation-ubiquitination signaling that regulates DNA repair factors accrual at damaged chromatin as long as there is a sufficient tail length for the MDC1 tandem BRCT domain to interact[33].

### H2AX Y142L partially restores 53BP1 IRIF in H2AX KO cells
Interestingly, in addition to the C-terminal linker region, the functional residue tyrosine 142 (Y142), which is required for MDC1 BRCT interaction, is evolved from leucine (L) in yeast or phenylalanine (F) in plants (Figs. 1b, 2d)[3]. As we demonstrated that the evolutionarily diverse linker region (a.a.120–137) (Supplementary Fig. 2a) does not play a functional role in γH2AX-MDC1-mediated ubiquitin signaling in recruiting 53BP1 to damaged chromatin (Fig. 1e), we then used the naturally occurring MDC1 binding-defective (Y142L) and -proficient (Y142F) evolutionary variants[33] to examine the function of the H2AX linker region in downstream DNA repair IRIF formation. As expected, we observed MDC1 IRIF formation in Y142F reconstituted H2AX KO cells (Supplementary Fig. 2b). Strikingly, a subset of Y142L reconstituted H2AX KO cells form 53BP1 IRIF despite inability to form MDC1 IRIF (Fig. 2e, h and Supplementary Fig. 2c), which is inconsistent with our current understanding of the genetic regulation of 53BP1 damaged chromatin recruitment that requires MDC1-RNF8-RNF68 mediated H2A(X) K15 ubiquitination signaling[9,19,20,30,34–37]. We also observed IRIF formation in the Y142L reconstituted H2AX KO cells for the 53BP1 downstream effector protein, RIF1 (Supplementary Fig. 2d–e), suggesting that the MDC1-independent 53BP1 DNA damage recruitment is functionally relevant.

Using the chimeric protein approach, we fused H2AZ and macroH2A with the SQEL motif at their C-termini (Fig. 2f). Unlike the H2AZ-SQEY and MacroH2A-SQEY chimeric proteins, H2AZ-SQEL and macroH2A-SQEL did not restore 53BP1 recruitment in H2AX KO cells (Fig. 2g, h and Supplementary Fig. 1e, f). We also confirmed that 53BP1 can form IRIF without MDC1 IRIF formation within a single cell in H2AX KO with Y142L reconstitution (Fig. 3a). Substitution of tyrosine 142 with valine (Y142V), another amino acid containing a hydrophobic side chain that does not alter H2AX S139 phosphorylation[32], also resulted in partial 53BP1 IRIF formation (Supplementary Fig. 2f–h)[38]. Notably, we do not observe BRCA1 foci formation in Y142L reconstituted H2AX KO cells (Supplementary Fig. 2i, j), suggesting that unlike 53BP1, BRCA1 recruitment relies on MDC1 recruitment to DNA damage.

### H2AX promotes 53BP1 damaged chromatin recruitment independent of MDC1 IRIF formation
Our data provide evidence of a unique molecular requirement on H2AX that promotes 53BP1 damaged chromatin recruitment, and that this regulation is independent of the Y142 residue and MDC1 IRIF formation. Since we could functionally separate them with the Y142 mutant, we reasoned that the H2AX phosphorylation and ubiquitination pathways can work both epistatically as well as independently. To test this, we ectopically expressed RNF8 or RNF168 in H2AX KO cells to elevate the ubiquitin signals on chromatin (Fig. 3b, c). Reciprocally, we also overexpressed H2AX in RNF168 KO cells that we generated previously[29]. In both genetic settings, 53BP1 was able to form IRIF in the absence of either H2AX S139 phosphorylation or RNF168-mediated ubiquitination (Fig. 3b). Notably, the GFP-expression level is heterogeneous in our KO-reconstitution system. Although we did observe that there is a general trend of GFP expression level-dependent 53BP1 foci formation in both wildtype and Y142L reconstituted KO cells, S139A mutant was not able to rescue 53BP1 recruitment in H2AX KO cells regardless of the expression level (Supplementary Fig. 3a). These data suggest that while the level of 53BP1 recruitment can be affected by the level of H2AX, which is consistent with the current model of H4K20me2 oscillation throughout the cell cycle and 53BP1 recruitment

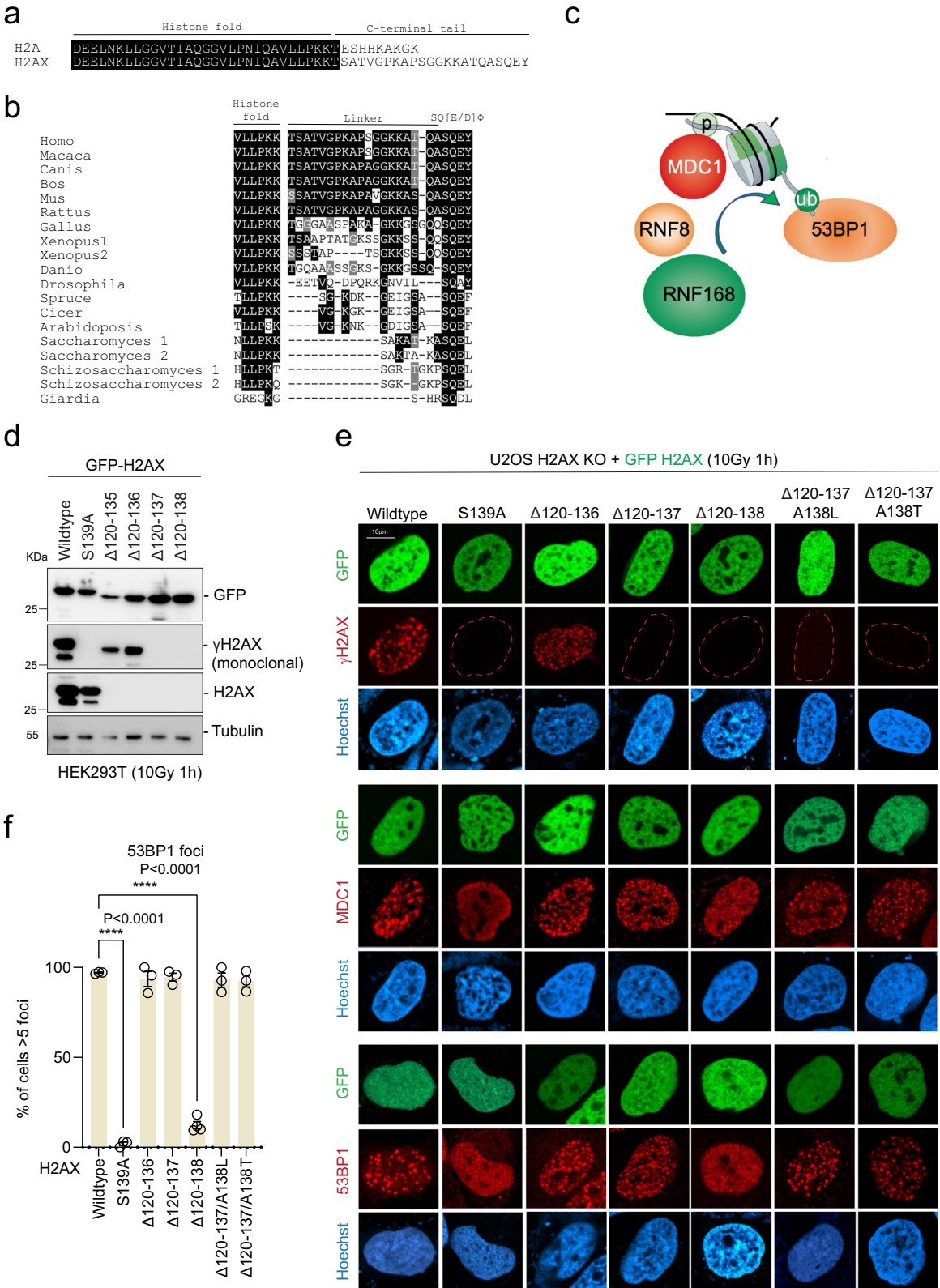

**Fig. 1 | The H2AX-(X)pSQEY is the minimal DDR activating motif. a** Sequence alignment of human H2A and H2AX. Conserved residues are highlighted in black. **b** Sequence alignment of H2AX homologs C-terminal tail sequence across different species. Conserved residues are highlighted in black. **c** Schematic illustration of the H2AX-mediated phosphorylation-ubiquitination DNA damage response pathway. **d** Immunoblots of H2AX and γH2AX antibodies for H2AX mutants. Repeated three times independently with similar results. **e** Representative immunofluorescence micrographs of ionizing radiation induced foci for γH2AX, MDC1 and 53BP1 in H2AX KO with GFP-H2AX deletion mutant reconstitution at 1 h after 10 Gy radiation. **f** Quantification of 53BP1 foci as represented in (**e**) for the indicated H2AX mutants. The error bars correspond to mean ± SD of three independent experiments. Two-tailed unpaired *T* test. Source data are provided as Source Data file.

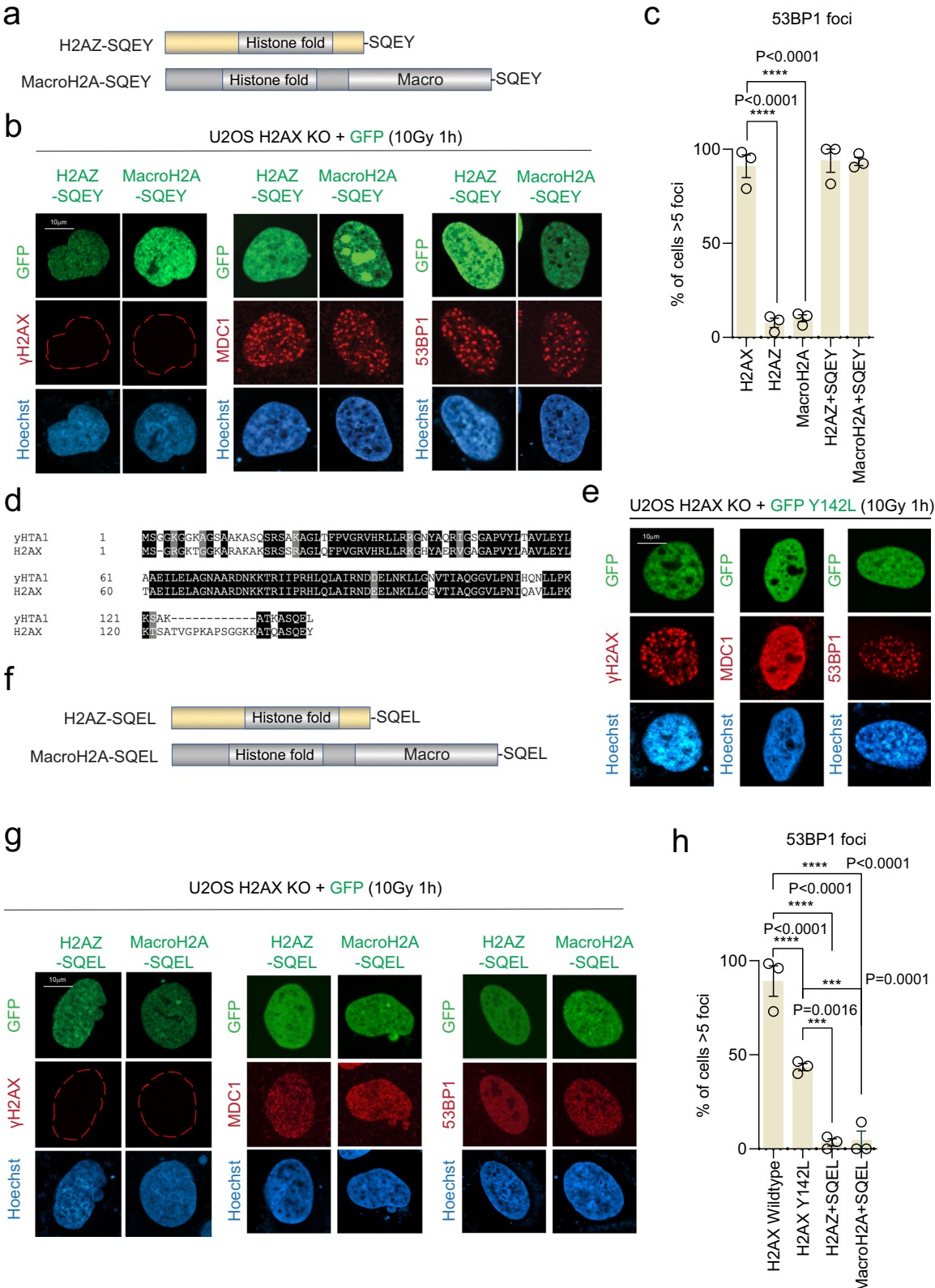

and physiological function[38–40], the genetic regulation is still the primary determining factor for 53BP1 recruitment. Overall, our data suggest that there is an alternative regulatory mechanism to recruit 53BP1 to damaged chromatin that is independent of MDC1 IRIF formation. More importantly, the 53BP1 IRIF formation in Y142L reconstituted H2AX KO cells is regulated by the distinctive H2AX C-terminal linker region.

### H2AX C-terminal linker region facilitates 53BP1 recruitment through its evolutionarily conserved GKK--Q residues

To further dissect the molecular genetics within the H2AX C-terminal tail for 53BP1 IRIF formation independent of the ubiquitin signaling, we use the yeast H2AX ortholog HTA1 (yHTA1) as the genetic backbone to reconstruct the molecular requirement for 53BP1 IRIF formation by humanizing yHTA1 to human H2AX. As expected, yHTA1 cannot form

**Fig. 2 | Histone H2A variants-SQEY chimera proteins activate the DDR pathway.**
**a** Schematic illustration of H2AZ and MacroH2A chimera constructs with the SQEY motif. **b** Representative immunofluorescence micrographs of ionizing radiation induced foci for γH2AX, MDC1, and 53BP1 in H2AX KO with H2AZ-SQEY or MacroH2A-SQEY chimera protein expression at 1 h after 10 Gy radiation. **c** Quantification of 53BP1 foci as represented in (**b**) and Supplementary Fig. 1d for the indicated constructs. The error bars correspond to mean ± SD of three-four independent experiments. Two-tailed unpaired *T* test. **d** Sequence alignment between yeast HTA1 (yHTA1) and human H2AX. **e** Representative

immunofluorescence micrographs of ionizing radiation induced foci for γH2AX, MDC1 and 53BP1 in H2AX KO with GFP-H2AX Y142L at 1 h after 10 Gy radiation. **f** Schematic illustration of H2AZ and macroH2A chimera proteins with the SQEL motif. **g** Representative immunofluorescence micrographs of γH2AX, MDC1 and 53BP1 staining in H2AX KO with GFP-H2AZ-SQEL or GFP-MacroH2A-SQEL at 1 h after 10 Gy radiation. **h** Quantification of 53BP1 foci as represented in (**e**) and (**g**) for the indicated expression vectors. The error bars correspond to mean ± SD of three independent experiments. Two-tailed unpaired *T* test. Source data are provided as Source Data file.

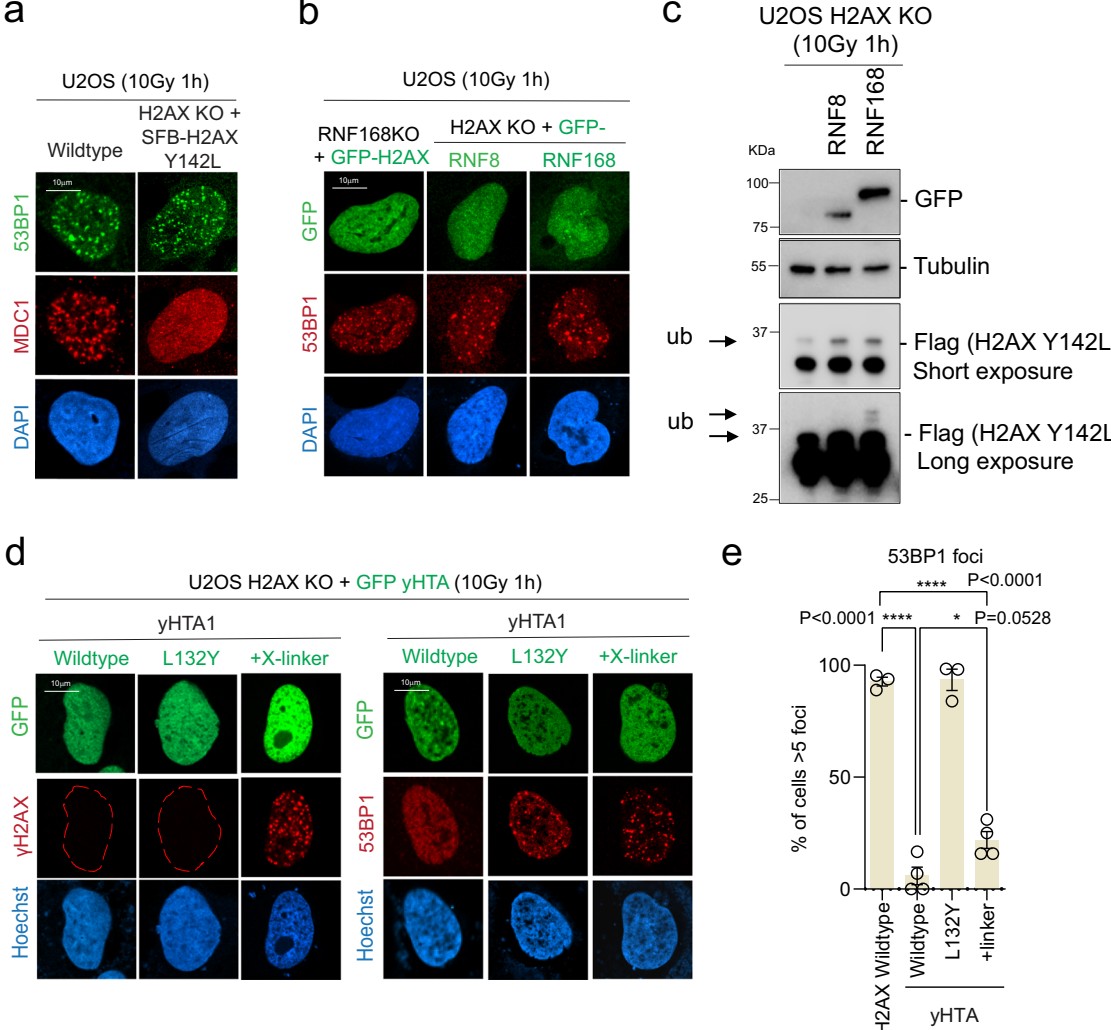

**Fig. 3 | H2AX C-terminal linker region promotes 53BP1 IRIF. a** Representative immunofluorescence micrographs for MDC1 and 53BP1 colocalization in U2OS wildtype and H2AX KO with SFB-H2AX Y142L reconstitution, at 1 h after 10 Gy radiation. Experiment was performed independently three times with similar results. **b** Representative immunofluorescence micrographs of 53BP1 in U2OS RNF168 KO with GFP-H2AX overexpression (left) and H2AX KO cells with GFP-RNF8 (middle) or GFP-RNF168 (right) at 1 h after 10 Gy radiation. Experiment was performed independently three times with similar results. **c** Immunoblot of SFB-H2AX Y142L in U2OS H2AX KO with or without RNF8 or RNF168 ectopic expression.

Experiment was performed independently three times with similar results. **d** Representative immunofluorescence micrographs of ionizing radiation-induced foci for γH2AX, MDC1, and 53BP1 in H2AX KO with GFP-yHTA, yHTA-L132Y, and GFP-yHTA+X-linker mutants using γH2AX, MDC1, or 53BP1 at 1 h after 10 Gy radiation. **e** Quantification of 53BP1 foci as represented in (**d**) for the indicated expression vectors. The error bars correspond to mean ± SD of three-four independent experiments. Two-tailed unpaired *T* test. Source data are provided as Source Data file.

53BP1 IRIF, but yHTA1 L132Y rescues 53BP1 IRIF formation in H2AX KO cells (Fig. 3d). We believe that the tyrosine residue restores the interaction with the MDC1 BRCT domains[19]. Consistent with our hypothesis that MDC1-independent 53BP1 recruitment is mediated through the H2AX C-terminal linker region, yHTA1 with the addition of human H2AX-linker (yHTA1+X-linker) was able to partially restore 53BP1 foci

(Fig. 3d, e and Supplementary Fig. 3b). Moreover, H2AZ-X-linker-SQEL was also able to restore 53BP1 IRIF similar to the H2AX Y142L mutant in H2AX KO cells (Supplementary Fig. 3c–e), suggesting that tyrosine Y142 has evolved to regulate MDC1-mediated downstream ubiquitination signaling, while the human H2AX linker region plays a role in promoting 53BP1 IRIF formation independent of MDC1 IRIF formation.

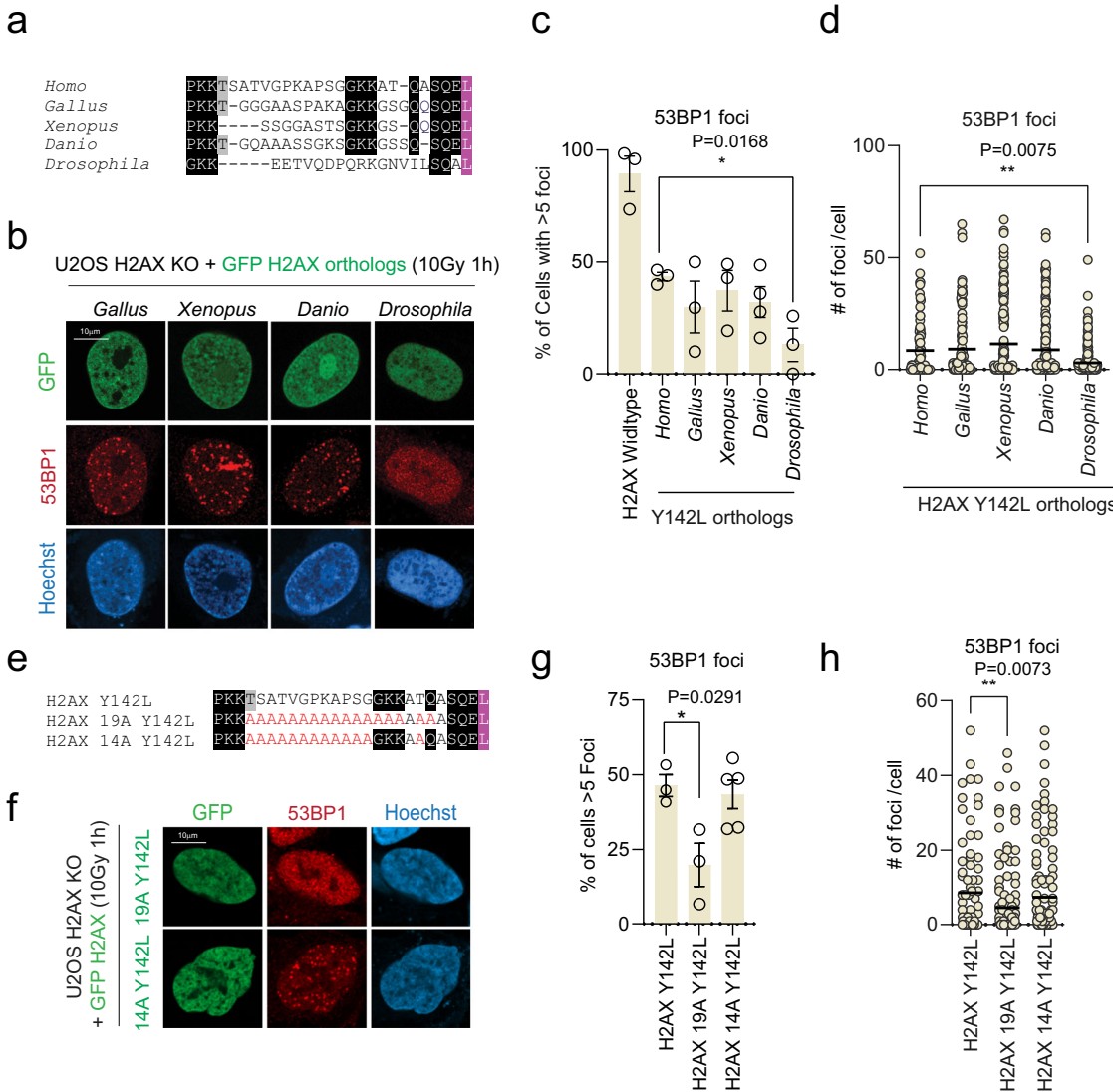

**Fig. 4 | Molecular dissection of the H2AX C-terminal linker region-mediated 53BP1 IRIF formation. a** Sequence of H2AX ortholog Y to L mutants. Conserved residues are highlighted in black. Leucine substitution highlighted in purple.
**b** Representative immunofluorescence micrographs of ionizing radiation-induced foci for 53BP1 in H2AX KO with GFP-H2AX ortholog Y to L mutant reconstitutions at 1 h after 10 Gy radiation. **c** Quantification of cells with >5 53BP1 foci as represented in (**b**) for the indicated expression vectors. The error bars correspond to mean±SD of three-four independent experiments. Two-tailed unpaired *T* test.
**d** Quantification of 53BP1 foci as represented in (**b**) for the indicated expression vectors. Foci counts are representative of ≥25 cells from three independent experiments and line is representative of the mean. Two-tailed unpaired *T* test.

**e** H2AX mutants C-terminal sequences used in (**f**). Residues with an alanine substitution are labeled in red. Conserved residues are highlighted in black. Leucine substitution highlighted in purple. **f** Representative immunofluorescence micrographs of 53BP1 in H2AX KO with GFP-H2AX 19A Y142L or GFP-H2AX 14A Y142L at 1 h after 10 Gy radiation. **g** Quantification of cells with >5 53BP1 foci as represented in (**f**). The error bars correspond to mean ± SD of three-five independent experiments. Two-tailed unpaired *T* test. **h** Quantification of 53BP1 foci as represented in (**f**). Foci counts are representative of ≥25 cells from three independent experiments and line is representative of the mean. Two-tailed unpaired *T* test. Source data are provided as Source Data file.

Since the H2AX linker region is evolutionarily diverse among vertebrates (Fig. 1b), we next tested whether the H2AX orthologs from *G. gallus*, *X. laevis*, *D. rerio*, and *D. melanogaster*, which are relatively diverse at the linker region as compared to human H2AX (Figs. 1b, 4a), can also function in human cells with the Y to L mutation. We found that *G. gallus*, *X. laevis*, *D. rerio*, but not *D. melanogaster* Y to L mutants were able to restore 53BP1 IRIF (Fig. 4b–d and Supplementary Fig. 3f), while their wildtype counterparts harboring the C-terminal tyrosine restored most of the 53BP1 foci formation in human H2AX KO cells (Supplementary Fig. 3g). Sequence alignment analysis of these constructs showed that G132, K133, K134, and Q137, are conserved in *G. gallus, X. laevis, and D.rerio*, but not *D. melanogaster* (Fig. 4a). Strikingly, the human H2AX linker region mutant retaining G132, K133, K134, and Q137 on the Y142L backbone (H2AX 14 A Y142L) restored

53BP1 IRIF formation comparable to the H2AX Y142L mutant (Fig. 4e–h and Supplementary Fig. 4a). Electrostatic potential analysis showed that the H2AX linker region is positively charged (Supplementary Fig. 4b) and reversing the charge of the H2AX linker region largely abolished the 53BP1 IRIF formation (Supplementary Fig. 4a, c–e). Together, these data suggest that the GKK--Q residues within the H2AX C-terminal linker region are important for recruiting 53BP1.

## 53BP1 Oligomerization and Tudor domains are required for the interaction with phosphorylated H2AX C-terminal tail

The genetic regulation of 53BP1 recruitment at damaged chromatin is well documented[30,34,41–46]. Notably, besides H4K20me2 and H2A/X K15ub histone marks, 53BP1 was previously shown to bind phosphorylated H2AX C-terminal tails via BRCT domains[47] and amino acid

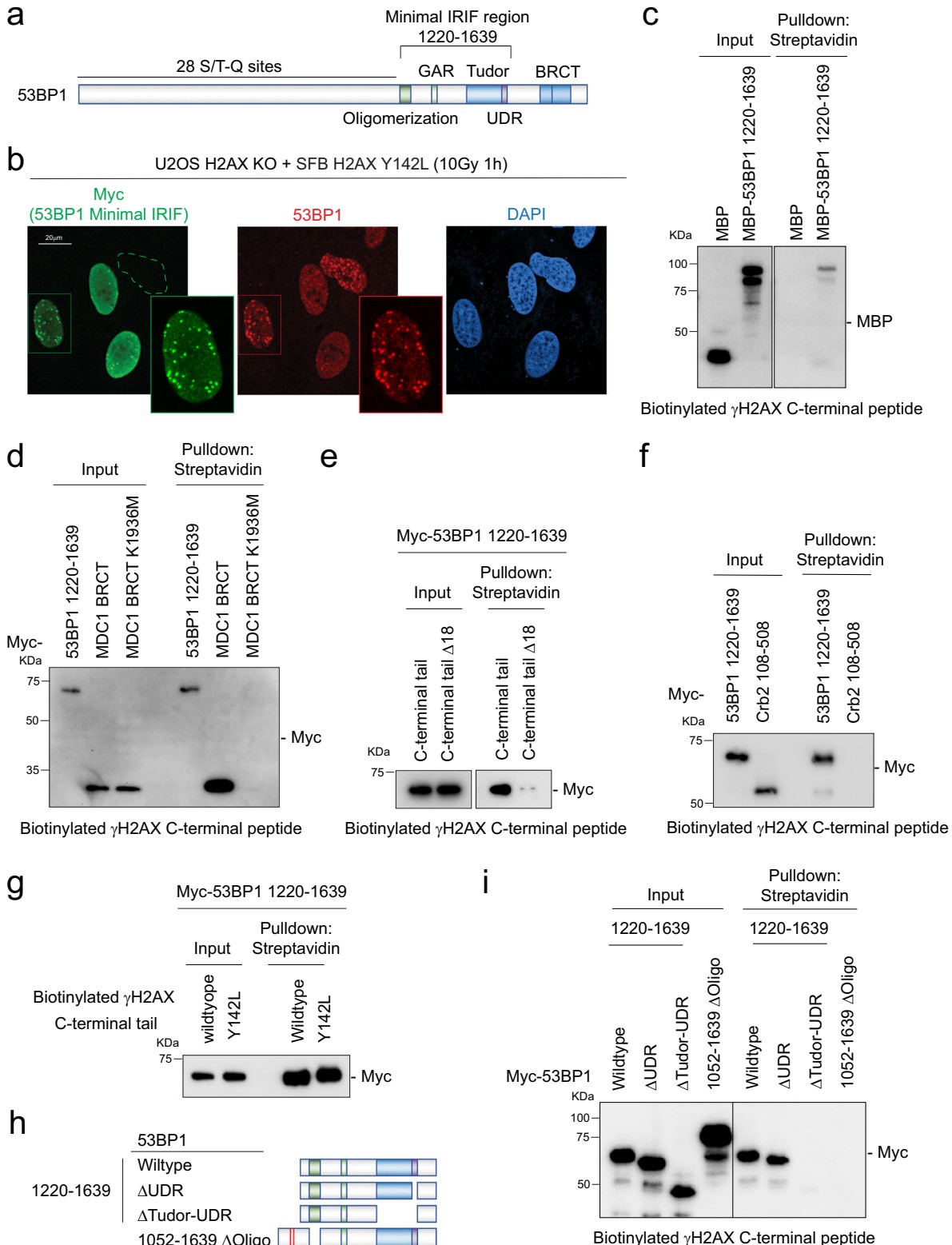

region 956-1354[48]. However, neither of these regions can form IRIF. To this end, we hypothesized that the minimal region for IRIF formation (a.a. 1220-1639) on 53BP1, which contains the oligomerization domain, tudor domain, and ubiquitin-dependent recruitment (UDR) motif (Fig. 5a)[30,45], is involved in H2AX linker-mediated 53BP1 IRIF formation. By immunofluorescence, we found that the myc-53BP1 a.a. 1220-1639 colocalized with endogenous 53BP1 in SFB-H2AX Y142L reconstituted

H2AX KO stable cells (Fig. 5b and Supplementary Fig. 4f). Since reversing the charge of the H2AX linker region dramatically reduced the 53BP1 IRIF formation (Supplementary Fig. 4c–e), we hypothesized that the γH2AX-linker mediated 53BP1 IRIF formation is through protein-protein interaction. Using a peptide pull-down assay, in vitro purified 53BP1 a.a. 1220–1639 showed specific binding affinity to a 23-amino acid phosphorylated H2AX C-terminal tail peptide (Fig. 5c). We

**Fig. 5 | H2AX C-terminal linker region interacts with 53BP1 via the oligomerization and tudor domains. a** Domain organization of 53BP1. Oligomerization, GAR, Tudor, UDR, and BRCT domains are indicated and the minimal IRIF forming region is denoted as containing the oligomerization, GAR, Tudor, and UDR domains. **b** Representative immunofluorescence images of Myc-53BP1 IRIF fragment (a.a.1220–1639) localization with endogenous 53BP1 in H2AX KO with stable reconstitution of SFB-H2AX Y142L at 1 h after 10 Gy radiation. Experiment was performed independently three times with similar results. **c** Pulldown assay with purified recombinant proteins MBP or MBP-53BP1 IRIF formation fragment a.a. 1220–1639 using biotinylated phosphorylated S139 H2AX C-terminal tail peptide. Repeated three times independently with similar results. **d** Pulldown assay with cell lysates expressing Myc-53BP1 IRIF formation fragment a.a. 1220-1639, MDC1 tandem BRCT domain or MDC1 BRCT domain with K1936M mutation using biotinylated phosphorylated S139 H2AX C-terminal tail peptide. Repeated three times independently with similar results. **e** Pulldown assay with cell lysates expressing

Myc-53BP1 IRIF formation fragment using biotinylated phosphorylated S139 H2AX C-terminal tail peptide with or without the 18 linker residues. Repeated three times independently with similar results. **f** Pulldown assay with cell lysates expressing Myc-53BP1 IRIF formation fragment or Myc-Crb2 a.a. 108–508 using biotinylated phosphorylated S139 H2AX C-terminal tail peptide. Repeated three times independently with similar results. Source data are provided as Source Data file. **g** Pulldown assay with cell lysates expressing Myc-53BP1 IRIF formation fragment using biotinylated phosphorylated S139 H2AX C-terminal tail peptide or biotinylated phosphorylated S139 H2AX C-terminal tail with Y142L mutation. Repeated three times independently with similar results. **h** Schematic diagram of 53BP1 fragments used in (**i**). **i** Pulldown assay of Myc-53BP1 IRIF wildtype or deletion mutants using biotinylated phosphorylated S139 H2AX C-terminal tail peptide. Repeated three times independently with similar results. Source data are provided as Source Data file.

also detected a specific interaction between 53BP1 and the phosphorylated H2AX C-terminal peptide in cells with moderately lower affinity than the MDC1 BRCT domains (Fig. 5d) and this interaction requires the H2AX linker region (Fig. 5e). Interestingly, the fragment harboring the oligomerization and tudor domains in Crb2, the fission yeast 53BP1 ortholog[49], did not show binding affinity to the γH2AX C-terminal tail (Fig. 5f). The binding affinity between 53BP1 and H2AX C-terminal tail did not show a discernible difference with H2AX C-terminal tail Y142L peptide (Fig. 5g), suggesting that the Y142L mutation itself does not affect 53P1 binding affinity to the H2AX C-terminal linker region. To further dissect the molecular interaction between 53BP1 and the H2AX C-terminal tail, we systematically generated deletion mutants for the peptide pulldown assay (Fig. 5h). Using cell lysates with overexpression of the different mutants, we found that both the tudor domain and oligomerization domain are required for the γH2AX-linker interaction (Fig. 5i). From an evolutionary perspective, the ubiquitin signaling regulators RNF8 and RNF168 are only present in higher eukaryotes, which require H2AX Y142 for MDC1-BRCT binding and subsequent activation of the RNF8-RNF168 ubiquitination cascade (Supplementary Fig. 5a). Alpha fold prediction also revealed that the higher eukaryotes have evolved a structural oligomerization domain that interacts with the H2AX C-terminal linker region (Supplementary Fig. 5b). Together, these data suggest that 53BP1 has an evolved molecular mechanism to bind to the H2AX linker region during recruitment to damaged chromatin.

### The H2AX C-terminal linker region-mediated 53BP1 IRIF formation is cell cycle-regulated

Nearly all of the H2AX wildtype cells were 53BP1 IRIF positive after IR treatment in contrast to the ~45% 53BP1 IRIF positive in the Y142L reconstituted H2AX KO cells. Interestingly, a subset of ionizing radiation-induced 53BP1 foci positive Y142L reconstituted H2AX KO cells displayed a comparable number of foci as wildtype (Fig. 6a). We reasoned that this unique pattern of H2AX linker region-mediated 53BP1 damage-induced foci is cell cycle-regulated. To test this, we used camptothecin (CPT), a topoisomerase I inhibitor, to induce DNA damage specifically in S-phase. Intriguingly, H2AX KO cells with Y142L reconstitution were not able to form CPT-induced 53BP1 foci (Fig. 6b–d). Consistent results were observed using low doses of radiation as well as CPT treatment (Supplementary Fig. 6a, b). These data suggest that H2AX linker region-mediated 53BP1 foci formation predominately occurs in the G1 phase.

Cell survival assays showed that the H2AX KO cells with Y142L reconstitution are more resistant to CPT treatment compared to the wildtype counterpart (Fig. 6e, f) indicating the absence of linker-mediated 53BP1 recruitment during the S-phase may tip the balance to enhance homologous recombination repair, which leads to higher tolerance to CPT-induced damage. Using the non-homologous end joining (NHEJ) reporter cell line (EJ5-GFP), we observed a significant

increase of distal end joining repair in cells with ectopic expression of H2AX wildtype, but only a modest increase in Y142L (Supplementary Fig. 6c). Ectopic expression of H2AX wildtype or Y142L mutant did not affect cell survival upon CPT treatment (Supplementary Fig. 6d). Consistent with overexpression, H2AX KO with Y142L reconstitution showed a lower distal end joining repair compared to H2AX KO with wildtype reconstitution (Supplementary Fig. 6e), potentially due to the inefficient 53BP1 recruitment to DNA damage sites. Interestingly, H2AX KO cells with reconstitution of wildtype and Y142L did not show a discernible difference in DR-GFP and EJ2-GFP reporter assays (Supplementary Fig. 6e), suggesting that these two mutants' function similarly in homologous recombination and alternative end-joining repair.

Using comet assay, we observed a reduced percentage of DNA in tails for H2AX KO cells reconstituted either H2AX wildtype and Y142L compared to H2AX KO cells, suggesting that there is less endogenous DNA damage. Percentage of DNA in tails increased in all three groups at 1 h after 2 Gy irradiation but only the H2AX wildtype or Y142L reconstituted cells could resolve the DNA damage after 24 h. These data suggest that the MDC1-independent 53BP1 recruitment is functionally relevant (Supplementary Fig. 6f). Similarly, using 53BP1 foci as readout, we observed damage foci clearance at longer timepoints (Supplementary Fig. 6g).

### Human H2AX and yeast H2A genes are not functionally interchangeable

In the budding yeast *S. cerevisiae*, there are two H2AX orthologs, *HTA1* and *HTA2*, which share 94% similar sequence homology but have evolved distinctive functions[50,51]. To study the evolutionary function of hH2AX C-terminal linker mutants in yeast, we generated *S. cerevisiae hta1Δ* and *hta2Δ* strains. No *hta1Δ hta2Δ* segregants were recovered, signifying that the double deletion is inviable. The wildtype segregant size was fully restored in cells bearing episomal *HTA1* expressed from a *GAPDH* promoter[52]. Expression of hH2AX in *hta1Δ HTA2* or *hta1Δ hta2Δ* cells failed to complement the slow-growth and inviability phenotypes, respectively.

We used a plasmid shuffle assay to further examine the ability of hH2AX C-terminal linker mutants to complement loss of yeast yH2A isoforms (Supplementary Fig. 7a). Cells are grown in medium that selects for both the plasmid bearing *HTA1* and the plasmid bearing the hH2AX or yH2A gene products, then spotted to plates that select against *HTA1*. Interestingly, in this case neither *hta1Δ HTA2* nor *hta1Δ hta2Δ* are able to grow after loss of the *HTA1* expressing plasmid (Supplementary Fig. 7b), suggesting that, in contrast to spores that can partially adapt to *HTA1* loss during germination, cycling cells require *HTA1*. This is consistent with experiments showing yeast cannot quickly adapt to new histone variants[53]. We found that hH2AX can complement growth in an *hta1Δ HTA2* single mutant, but not in an *hta1Δ hta2Δ* double mutant (Supplementary Fig. 7b). Further, none of

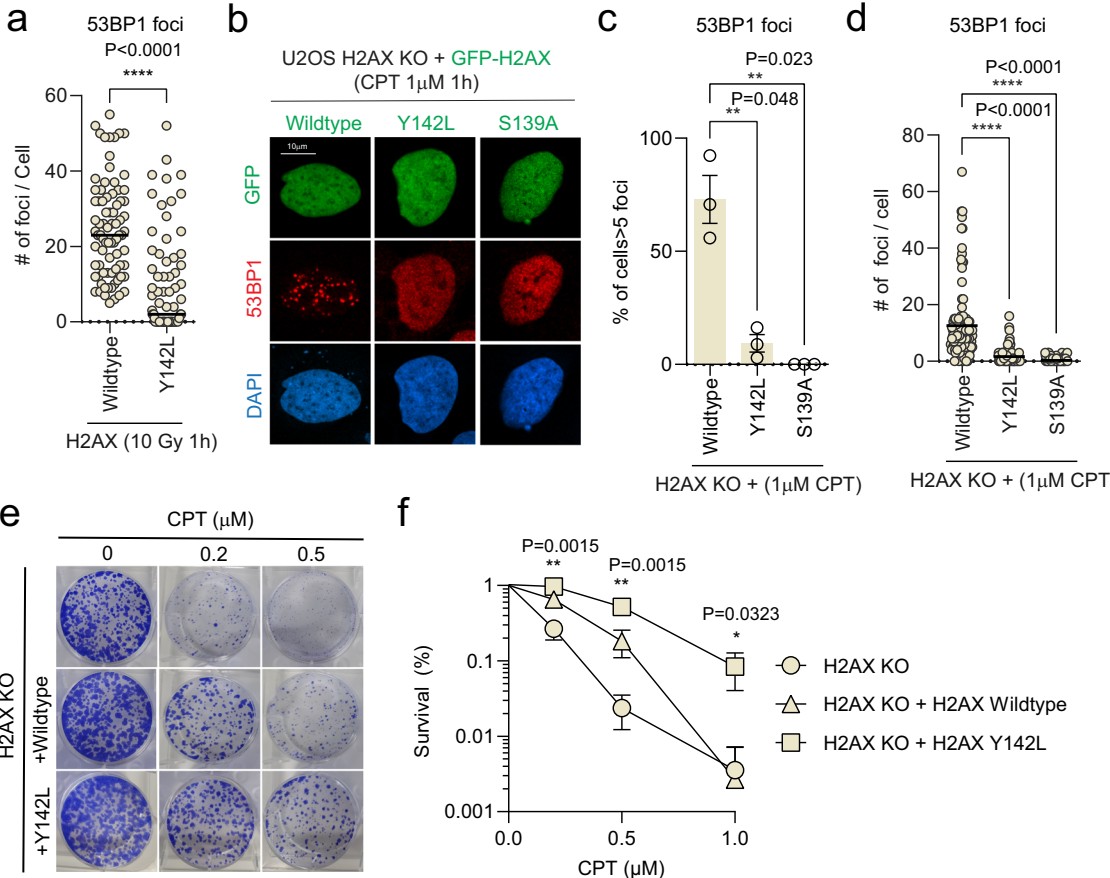

**Fig. 6 | The H2AX C-terminal linker region-mediated 53BP1 IRIF formation is cell cycle regulated. a** Quantification of the total number of foci per cell in H2AX KO cells expressing H2AX wildtype or Y142L 1 h after 10 Gy radiation. Data represents ≥25 cells from three independent experiments and the line is representative of the mean. Two-tailed unpaired $T$ test. **b** Representative immunofluorescence images for 53BP1 in H2AX KO cells with GFP-H2AX wildtype or mutant reconstitution, 1 h after 1 μM camptothecin treatment. **c** Quantification of cells with >5 53BP1 foci in H2AX KO cells expressing GFP-H2AX and mutants as shown in (**b**). The error bars correspond to mean ± SD of three independent experiments. Two-tailed unpaired $T$ test. **d** Quantification of the total number of 53BP1 foci per cell in H2AX KO cells with reconstitution of GFP-H2AX wildtype or mutant reconstitution 1 h after 1 μM camptothecin treatment as shown in (**b**). Data represents ≥25 cells from three independent experiments and the line is representative of the mean. Two-tailed unpaired $T$ test. **e** Representative images of clonogenic survival assay of H2AX KO cells with or without stable expression of SFB-H2AX wildtype or Y142L and treated with indicated doses of camptothecin. **f** Quantification of cell survival as represented in (**e**). The error bars correspond to mean ± SD of three independent experiments. Source data are provided as Source Data file.

the hH2AX variants tested complemented the hta1Δ hta2Δ double mutants, and we found that they also failed to fully complement in the hta1Δ HTA2 background, as clear growth defects were observed. Cells were treated with 3 mM camptothecin (CPT) to determine response to DNA damaging agent. As expected, mutation of S129 in yeast sensitized the cells to CPT, particularly in the hta1Δ hta2Δ mutant, suggesting Hta2 phosphorylation can partially compensate for the lack of Hta1 phosphorylation. Humanizing yHTA1 at the C-terminus by adding the hH2AX linker and K127Q mutation resulted in only a slight defect in untreated and CPT-treated cells.

Overall, these results show that human H2AX is incompatible with yeast, potentially due to the lack of a function ortholog protein, such as MDC1[54], or H2AX-mediated DDR in higher eukaryotes have evolved beyond the capacity to be effective in yeast and that full loss of both yeast H2A isoforms results in inviable cells[53,54].

## Discussion

The roles of histone variant H2AX, particularly the damaged induced phosphorylation (γH2AX) has been well-characterized in the context of the DDR pathway. Considering the importance of H2AX in maintaining genome stability across different organisms, their N-terminal and C-terminal tails are not very well conserved. In this study, using DNA repair protein IRIF as a biological readout, we defined the minimal

requirement for the activation of the phosphorylation-ubiquitination pathway that is crucial for the recruitment of DNA repair proteins at DNA breaks. We also discovered a mechanism that promotes 53BP1 accumulation at damaged chromatin via the evolved H2AX C-terminal tail region between the histone fold domain and the -SQEY motif (linker region) and that is MDC1 IRIF independent. Notably, this non-canonical regulation of 53BP1 at damaged chromatin is mediated by interaction between the γH2AX-linker region and the 53BP1 oligomerization-tudor domains. Alphafold predictions of both Crb2 and Rad9 suggest the absence of secondary structures in the oligomerization domain (Supplementary Fig. 5b), which is required for the γH2AX-linker region interaction. We speculate that the H2AX C-terminal linker region and 53BP1 oligomerization domains co-evolved for a new epigenetic regulation for 53BP1 function at damaged chromatin in vertebrates. This is consistent with our data showing that the Crb2 fragment harboring the previously reported oligomerization-tudor domain does not interact with the γH2AX-linker region (Fig. 5f)[49]. Additionally, the critical residues for MDC1 IRIF-independent 53BP1 recruitment (GKK--Q) arose in and are conserved in vertebrates, similar to the evolved 53BP1 oligomerization domain. Throughout evolution, the very last residue of H2AX orthologs has evolved from leucine (in yeast) to phenylalanine (in plants) and to tyrosine (in vertebrates) (Fig. 1b). Seemingly, the functional H2AX Y142

binding protein, MDC1, the ubiquitin signaling pathway regulators (RNF8 and RNF168), the 53BP1 ubiquitin-dependent region (UDR), and the specific sequence of the H2A(X) N-terminal K15ub, were only found in vertebrates, suggesting that the higher eukaryotes have developed a more intricate DDR pathway involving multiple regulatory proteins to ensure spatial-temporal control of DNA repair proteins at damaged chromatin.

In yeast, *HTA1* and *HTA2* have a one residue difference at the C-terminus, contributing to additional roles for *HTA1* in the yeast DDR and marking the beginning of the evolution of H2AX as a specialized DDR histone variant[51]. The highly conserved phosphorylatable serine residue retains function in both HTA1 and HTA2 and is shown to support the yeast DDR by interacting with the 53BP1 homolog's (Crb2 or Rad9) tandem BRCT domains[55–58] in conjunction with the evolutionarily conserved H4K20me2 recognition by the tudor domain[46]. This is in contrast to human cells, which rely on MDC1-BRCT-γH2AX interaction for initiation of the DDR. Although the MDC1 ortholog, mdb1, was identified in *S. pombe*[59], exactly how it plays a role in regulating the DDR in yeast is not fully understood. Further, that there is no MDC1 homolog in *S. cerevisiae*, and lack of sequence conservation between the tyrosine residues in the yeast H2AX homologs which are critical for MDC1 interaction suggests a point of evolutionary divergence in DDR. Additionally, RNF8-RNF168 ubiquitin ligase orthologs have not been identified in yeast, and the yeast 53BP1 ortholog does not have the UDR motif, implicating that ubiquitination is an evolved signaling pathway for the DDR in humans[30].

Functionally, γH2AX linker-mediated 53BP1 regulation plays a different role from the canonical MDC1-dependent pathway. In our H2AX KO system, we observed almost all H2AX wildtype-reconstituted cells displayed 53BP1 IRIF while only a subset of H2AX Y142L-reconstituted cells rescued 53BP1 IRIF, representing γH2AX linker-mediated 53BP1 recruitment (Fig. 2h). We found that γH2AX linker-mediated 53BP1 regulation does not restore 53BP1 foci in H2AX KO cells during S-phase (Fig. 6b–d) when H4K20me2 is diluted from newly incorporated histones[60–62]. We speculate that γH2AX linker-mediated 53BP1 recruitment requires and cooperates with a higher level of H4K20me2 to achieve sufficient affinity for 53BP1 to engage at damaged chromatin in the absence of ubiquitin signaling. This is similar to our current understanding of the DNA damage-dependent RNF168-mediated H2A K15ub and cell cycle-regulated H4K20 methylation in orchestrating the choice of BARD1-BRCA1 or 53BP1 recruitment at DNA breaks[63].

The human 53BP1 tandem BRCT domains have been shown to bind to γH2AX in vitro[47,64]. However, 53BP1 BRCT domains alone are not sufficient to form IRIF, and deletion of the 53BP1 BRCT domains does not cause overt DDR defects[43,45,65]. Their interaction may mediate other physiological functions in a context-specific manner, such as heterochromatin repair[64]. Moreover, a recent structural study using chemically synthetic nucleosomes harboring all three modifications did not observe any BRCT domain and H2AX C-terminus interaction[66], raising the question of whether their interaction is relatively transient or requires an additional interface for a stable interaction. Further investigation including the oligomerization domain will help elucidate the dynamics of the trivalent interactions between 53BP1 and modified nucleosome.

The evolutionary advantage and physiological significance of the γH2AX linker regulated 53BP1 recruitment is still not entirely clear. The Y142 is constitutively phosphorylated by WSTF[67] and transiently de-phosphorylated by EYA1 upon DNA damage, which facilitates MDC1 binding with γH2AX[68]. During the repair process, H2AX Y142 is re-phosphorylated which limits MDC1 binding. In our finding, we speculate that this epigenetic regulation helps retain 53BP at damaged chromatin when Y142 gets re-phosphorylated and MDC1 binding is reduced after DNA damage[68]. Since this epigenetic regulation may promote resistance to CPT, inhibition of this

regulation is potential target for sensitizing cells to DNA damaging agents.

## Methods

### Cell lines
U2OS cell line was obtained from American Type Culture Collection (ATCC). U2OS and RPE1 H2AX KO cells were obtained from Dr. Steven Jackson's lab at Gurdon Institute[69]. U2OS cell lines were maintained in Dulbecco's modified Eagle medium supplemented with 10% fetal bovine serum,100 U/mL penicillin, and 100 μg/mL streptomycin at 37 °C and 5% CO2. RPE-1 cells were maintained in DMEM:F21 supplemented with 10% fetal bovine serum, 100 U/mL penicillin, 100 μg/mL streptomycin, 2mM L-glutamine, and 0.25% sodium bicarbonate at 37 °C and 5% $CO_2$.

### Antibodies
Primary antibodies used in this study were rabbit polyclonal GFP (Invitrogen, A11122), mouse monoclonal γH2AX (JBW301) (1:1000 for western blot and immunofluorescence. EMD Millipore, 05-636), mouse monoclonal Flag (M2) (1:4000 for western blot and immunofluorescence. Sigma, F1804), mouse monoclonal c-myc (9E10) (1:1000 for western blot. Santa Cruz sc-40), rabbit monoclonal MBP (1:2000 for western blot. EPR4744) (Abcam ab119994), mouse monoclonal beta tubulin (1: 5000 for western blot. Santa Cruz, sc-166729), mouse monoclonal BRAC1 (1: 50 for immunofluorescence. Santa Cruz, sc-6954), mouse monoclonal RIF1 (1:100 for immunofluorescence. Santa Cruz, sc-515573). For western blot, secondary antibodies HRP-linked anti-mouse IgG (1: 2000. 115-035-166) and HRP-linked anti-rabbit IgG (1:2000. 115-035-144) were purchased from Jackson ImmunoResearch Laboratories. For immunofluorescence studies, Alexa Fluor 594 goat anti-mouse (1:500. Invitrogen, A32742) and Alexa Fluor 488 goat anti-rabbit (1:500. Invitrogen, A32731) were used. DAPI (1:5000. Fisher Scientific, D1306) was used for nuclear staining. For western blot, primary antibodies were incubated overnight at 4 °C and secondary antibodies were incubated at room temperature for 1 h. For immunofluorescence, primary antibodies and secondary antibodies were incubated in a humidified chamber at room temperature for 1 h.

### Plasmids and cloning
Human H2AX and yeast yHTA expression vectors were acquired as previously described[29,70]. H2AX C-terminal deletions, yHTA point mutations, H2A variants-SQEY and -SQEL fusion genes, and 53BP1 fragments and deletions were cloned by PCR using specific primers. H2AX homologs for different species and mutants were synthesized as gene blocks containing Gateway compatible recombination sequences and subcloned into the gateway entry vector p201 according to the Gateway cloning system protocol (ThermoFisher). 53BP1 expression vector (a.a. 1220–1639 and a.a. 1052–1639) were used as previously described[71]. H2AX and 53BP1 mutants were either generated using Q5 site-directed mutagenesis kit (New England Biolabs) or synthesized as Geneblock (Integrated DNA Technologies) and subcloned in the p201 using gateway system. All p201 were verified by sequencing in house and subcloned into expression vectors pMH-SFB-pDEST, Myc-pDEST, EmGFP-pDEST and MBP-pDEST using the gateway cloning system. For yeast studies, HTA1 and H2AX sequences were cloned into destination vectors pAG415GPD-ccdB or pAG416GPD-ccdB (kind gifts from Susan Lindquist) using gateway cloning system. Sequences for primers and Geneblocks are provided as Supplementary Data 1.

### Sequence alignment and molecular graphics
Sequence alignment was performed using Clustal Omega and box-shade. Structural molecular graphics were generated using PyMOL. Electrostatic potential was analyzed by Adaptive Poisson-Boltzmann Solver and the AMBER force field. Residues conservation was calculated using ConSurf server (https://hpc-status.tau.ac.il).

## Protein purification

MBP-53BP1 construct was transformed into BL21 cells grown overnight at 37 °C. Protein expression was induced by 1 μM IPTG and harvested in NETN (100 mM NaCl, 20 mM Tris-Cl pH 8.0, 0.5 mM EDTA, 0.5% (v/v) Nonidet P-40 (NP40) alternative with protease inhibitors) + 1% triton-X by sonication. and sonicated at 20% amplitude for 10 s on, 20 s off for a total of 2 min of sonication, then rotated for 1 h at 4 °C followed by centrifugation at 21,000 $g$ for 10 min at 4 °C. Supernatant was then incubated with amylose resin for 1 h with rotation at 4 °C to bind MBP-tagged protein. Resin was then washed three times with NETN buffer and protein was eluted using maltose elution buffer (200 mM NaCl, 20 mM Tris-HCl, 1 mM EDTA, 1 mM DTT) with 10 mM maltose.

## Peptide pulldown assay

H2AX peptides with the sequence N-biotin-TSATVGPKAPSGGKKAT-QApSQEY, N-biotin-TSATVGPKAPSGGKKATQApSQEL and N-biotin-ApSQEY were obtained from genscript resuspended at 4 mg/mL in distilled water. For pulldown assay using HEK293T cell lysates, 10 μg myc-tagged expression vectors were transfected into 10 cm plates. Cell lysates were harvested in NETN buffer and incubated with 2.5 μg of peptide and 10 μL streptavidin beads for one h at 4 °C with rotation. Beads were then washed for three times with NETN and resuspended in loading buffer for western blot analysis. For in vitro pulldown assay, MBP-purified protein was incubated with 10.5 μg of biotinylated peptide and 10 μL of streptavidin beads for 1 h at 4 °C. The beads were then washed for three times with NETN and resuspended in SDS-PAGE loading buffer for western blot analysis.

## Western blotting analysis

Lysates were loaded onto 10% Tris-Glycine gels for electrophoresis followed by transfer onto a PVDF membrane. Membranes were then briefly washed with tris-buffered saline containing 0.1% Tween (TBS-T) and blocked with 5% milk in TBS-T for 10 min followed by three 5-min washes in TBS-T. Membranes were then probed with the primary antibodies (3% BSA with sodium azide) overnight at 4 °C. The blots were then washed with (TBS-T) three times for 5 min each followed by incubation with HRP-conjugated secondary antibodies for 30 min. Blots were washed three times in TBS-T for 5 min each and then developed using ECL chemiluminescent substrate (Thermo Fisher) and imaged on a ChemiDoc MP (BioRad).

## Immunofluorescence

Cells were transfected using 3 μg plasmid and 10 μg PEI and seeded onto coverslips. After 24 h, coverslips were subjected to 10 Gy irradiation followed by 1 h of rest in the 37 °C incubator. The coverslips were then fixed with 3% paraformaldehyde and permeabilized with 0.5% Triton-X in PBS. For MDC1 immuno-staining, coverslips were pre-extracted with 0.5% Triton plus sucrose in PBS followed by fixation. Samples were incubated with primary antibodies for γH2AX, 53BP1, and MDC1 followed by Alexa Fluor 594 goat anti-mouse or Alexa Fluor 488 goat anti-rabbit secondary antibodies. The nucleus was stained using DAPI (1:5000) or Hoechst (1:2000). Samples were mounted with 0.02% anti-fade solution (0.02%, in 90% glycerol in PBS). Samples were imaged using a Nikon Ti2 C2+ confocal microscope with NIS elements software or Zeiss LSM900 confocal microscope. Images were analyzed using ImageJ and data analysis were done using GraphPad Prism (GraphPad Software Inc). At least 50 cells were analyzed in each experiment. Ionizing radiation-induced foci were defined as distinct puncta formed within the nucleus compared with non-irradiated group.

## Clonogenic survival assay

Cells were plated at a density of 1000 per well of a 6 well plate in triplicate and let rest for 24 h prior to treatment. Cells were treated with camptothecin for 1 h. After 14 days, plates were fixed/stained in Coomassie blue and counted for analysis.

## Cell-Titer-Glo

Cells were plated at a density of 2000 per well in a 96-well plate in triplicate and let rest for 6 h prior to treatment. Cells were treated with camptothecin for 1 h, followed by replacement with fresh media. After 5 days, CellTiter-Glo reagent was mixed 1:1 in the plate to lyse the cells. After 10 min, 50 μL of supernatant was plated into a new black-bottomed 96-well plate and read on a BioXCell Plate reader for luminescence at 547 nm.

## Comet assay

Alkaline comet assay was performed per the manufacturer's instructions (Biotechne). 5000 cells were immobilized into low-melt agarose and seeded onto CometSlides (Biotechne). Cells were then lysed for 1 h followed by treatment with alkaline unwinding solution. DNA was then run on gel electrophoresis under alkaline conditions and on ice for 30 min at 21 V. Slides were then washed and fixed in 70% ethanol and stained using Sybr Gold. Comets were imaged using an EVOS FL Auto and scored using CometScore Pro (Tritek).

## Repair pathway choice reporter assays

DNA repair pathway choice reporter assays were performed as previously described[72]. Briefly, EJ5-GFP cells were transfected with or without pCBASecI and SFB-H2AX wildtype or Y142L plasmids by electroporation. U2OS H2AX KO cells with SFB-H2AX and Y142L stable expression were transfected with reporter vectors of DR-GFP, EJ5-GFP, and EJ2-GFP and pCBASecI by electroporation. GFP-positive cells were quantified by flow cytometry 48 h after transfection.

## Yeast strains

All *S. cerevisiae* strains are isogenic to the W303 background[73] except that they carry a wild-type *RAD5* allele. Yeast genetic and molecular techniques were employed as described previously[74]. The *hta1::HPH-MX* and *hta2::KAN-MX* alleles were generated by fragment insertion. Plasmids and linear DNA fragments were transformed via lithium acetate[75]. For the plasmid shuffle assay, single colonies were grown in a medium lacking uracil and leucine and grown for 24 h at 30 °C. Cultures were diluted to 5 OD/mL and 5 μL 10-fold dilutions were spotted to plates lacking leucine and containing 5-fluoroorotic acid. Plates were incubated at 30 °C for 3 days.

## Statistical analysis

Data are represented as Mean ± S.D. as indicated from at least three independent biological replicates. Analyses using GraphPad Prism by ANOVA and Student's t-test. Significance was reported starting at $p < 0.05$.

## Reporting summary

Further information on research design is available in the Nature Portfolio Reporting Summary linked to this article.

# Data availability

All data supporting the findings of this study are available within the paper and its Supplementary Information. Source data are provided with this paper.

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

## Acknowledgements

The authors would like to thank Dr. James Haber at Brandeis University for insightful discussion. We also thank Drs. Fen Xia and Alicia Byrd at UAMS for technical advice, Drs. Davis Gius, Sang-Eun Lee, and Feng-chun Yang at UTHSCSA for sharing equipment and Dr. Stephen Jackson at the University of Cambridge for sharing H2AX knockout cell lines. J.W.L is supported by grants from NIH (NIGMS: R35GM137798, NCI: R01CA244261), American Cancer Society (RSG-20-131-01-DMC and TLC-21-164-01-TLC). University of Texas STARs award. E.D. is supported by a grant from NIEHS (1R21ES035997-01). W.Z. is supported by grants from NIH (NIGMS: R01GM141091, NCI: R01CA268641) and the American Cancer Society (RSG-22721675-01-DMC). N.R.P. is supported by grants from the American Cancer Society (IRG-16-187-13) and the National Cancer Institute (R37CA266042 and R01CA276470). M.L.F. is supported by an NCI Diversity Supplement. B.K. is supported by grants from the NIH (OD: DP5OD031863 and NIAID: R21AI173759).

## Author contributions

J.K., N.P., and J.L. conceived and designed the study. J.K., M.F., K.S., K.T., W.S., C.S., S.L., E.D., W.Z., N.P., and J.L. performed the experiments and analyzed the data. J.K., N.P., and J.L. wrote the original draft and E.D., W.Z., B.K., K.T., and W.S. reviewed and edited the paper with input from other authors.

## Competing interests

The authors declare no competing interests.
