## [Peer Review File · Nature Communications]

Evolved histone tail regulates 53BP1 recruitment at damaged chromatinREVIEWER COMMENTS

Reviewer #1 (Remarks to the Author):

γH2AX is the central player in DNA damage response and critical to sustain genome stability. Phosphorylation of γH2AX at Serine 139 has been widely accepted and used as a hallmark for damaged DNA. In this manuscript, Kelliher and colleagues present interesting results on the function of other surrounding γH2AX residues through the evolutionary perspective. After defining with the minimal DDR motif SQEY, Kelliher and colleagues found the Y142 is evolutionally divergent. Functionally, they found the Y142L is capable to recruit 53BP1 at the damaged chromatin independent from the MDC1 pathway. While the overall finding is new and might shed new light on how DNA repair signaling is activated across species, this reviewer feels that several important questions/experiments should be addressed to strengthen the conclusion before publication.

1 The author clearly demonstrated that overexpression of Y142L can induce 53BP1 foci. However, the downstream effectors like RIF1 after 53BP1 foci formation is not determined and this should be done to strengthen the conclusion.

2 In Figure 1b, the authors show F142 in Arabidopsis. Plants have different response to UV exposure and this F142 might explain why plants can survive the continuous UV exposure. This reviewer feel it is necessary to examine the 138F function in the same setting.

3 Does Y142L activate the BRCA1 pathway? A quick measure of BRCA1 foci formation should be done.

4 The 53BP1 formation after Y142L expression indicates the activation of NHEJ pathway. This can be determined by using the reporter assay after transient expression of Y142L mutant.

5 The evolution of SQEY to activate DDR suggests this motif might be important during cancer development. Is any mutation identified within the SQEY region from the TCGA database? This might provide additional information about how SQEY evolution can reshape the DNA damage signal in cancer.

6 Does Y142L overexpression change the drug sensitivity? Is any genome abnormality detected in Y142L overexpressed cells?

7 The data from polyclonal antibody in figure 1d is a little confusing and is better removed.

8 The title "Evolved histone tail regulates 53BP1 damaged chromatin recruitment" of the paper is confusing. This one might be more straightforward: "Evolved histone tail regulates 53BP1 recruitment at damaged chromatin".

9 Since figure 2 has only 3 panels, it can be incorporated/combined with figure 3.

Reviewer #2 (Remarks to the Author):

Phosphorylation of H2AX is an important early step in the signaling of DNA damage, stimulating the recruitment of DNA damage repair proteins. MDC1 directly binds to and recognizes γH2AX, which in turn leads to recruitment of the RNF8 and RNF168 ubiquitin E3 ligases, the subsequent ubiquitination of H2A and the binding of 53BP1 or BRCA1/BARD1, to coordinate repair. Kelliher et al. characterizes the C-terminal tail of H2AX and describes the minimal motif necessary to trigger the DNA damage

response. In addition, the authors find recruitment of 53BP1 to ionizing radiation induced foci that is independent on MDC1 binding. These findings are of potential interest for the entire DDR field.

Kleiner et al (Ref.61) already demonstrated that 53BP1 can bind gH2AX in the absence of MDC1. This reference should be cited much earlier in the manuscript and kept into account when drawing conclusions and providing novelty statements. The work in the submitted manuscript is still valuable in understanding this interaction at higher resolution, as they use different H2AX tail mutants. However, due to the comments below, there are concerns that needs to be addressed before publication.

Functional assays to test the role of 53BP1 recruitment in the absence of MDC1 needs to be included. Does H2AK15Ub still occur (check by western blot)? Is it possible that this mark is placed independently on MDC1, explaining how 53BP1 is recruited? Are there other factors recruited at these sites? Does the linker-mediated 53BP1 binding in absence of MDC1 actually still stimulates NHEJ? What are the effects on longer timepoints? Any evidence of this interaction being important for DNA damage/repair pathways would really help. For example, are the mutant-complemented H2AX +/- cells able to resolve damage?

This is important for the interpretation of the data. For example: lines 318-323: “..... suggesting that gH2AX linker-mediated 53BP1 recruitment requires and cooperates with a higher level of H4K20Me2 to achieve sufficient affinity...”. This seems too speculative without functional data.

Concerns on the experimental setup and interpretations:

- o The authors need to prove the presence of 53BP1 and lack of MDC1 on H2AX mutants, in the same IRIF co-localization experiment. Currently a lot of conclusions are drawn by separate experiments where only 1-2 proteins are stained (and there is not always a reference for foci formation, e.g. as in Figure 3B MDC1 staining). On this line, it is not clear how the authors define a foci in their image analysis. For example, in Fig 3D: MDC1 staining in H2AZ-SQEL looks like foci, and perhaps 53BP1 staining in macroH2A-SQEL as well. And in Fig 3B: Please include an image of a cell that doesn't have the 53BP1 foci or give us a bigger view with several cells, with and without foci. Including internal control or reference staining for foci helps to address these issues.
- o Are overexpression levels of H2AX mutants always the same? This needs to be quantified, as the authors show in figure 3G that overexpression of H2AX can by itself already induce the formation of foci. Therefore, the presence of foci can be dependent on the level of H2AX expression and not per se the mutation. It would be useful to bin foci formation based on GFP levels for the different H2AX mutants.
- o 10Gy and 1 mM CPT are very high doses for inducing damage. Is the 53BP1 effect also present at lower levels of damage?
- o Many images have overexposure as well as strange histones or DNA staining. For example, figure 2B and 3D, both left panels have DAPI staining that is starkly different than expected. Figure 2B, middle panel, macroH2A is enriched in the nucleolus, which indicates it isn't in chromatin properly and therefore shouldn't be used to draw any conclusions from. Figure 4G, GFP signal is overexposed.
- o Figure 4E: A control experiment with the WT H2AX sequences of these organisms is needed to show that WT fully rescues the effect. It is not possible to conclude with these data if the effect is due to the YtoL mutation or due to species differences.
- o Figure 5D-H: The authors need to include a control with the gH2AX Y142L (and/or other) mutant peptide, to confirm the dependency of 53BP1 binding on the motif the authors have identified.
- o The authors include a part about the evolution of the H2AX tail, by performing complementation experiments in yeast. This part is hard to follow and confusing. The effects of the mutants are really small and the conclusions of the authors on the presence or absence of a difference do not clearly match the colony growth results.

Concerns with data presentation:

- o It looks like data is reused for several figures. The WT and the Y142L look exactly the same in each graph (for i.e. % cells >5 53BP1 foci).
- o Supp fig 2D: This only shows two cells that were transfected, please add a quantification of more

cells. Same for Supp Fig 3C.

o Line 139-141: " We ectopically expressed RNF8 or RNF168 in H2AX KO cells to elevate the ubiquitin signals on chromatin". Please demonstrate that there are indeed increased ubiquitin signals on chromatin in these cells.

o Line 155: " yHTA1 L132Y rescues both MDC1 and 53BP1 IRIF formation.....(Figure 4a,b)...." There is no data showing the rescue of the MDC1 in these conditions. Please include it or edit the statement.

The method section needs to be expanded with the following details:

o Antibody dilutions for IF and incubation time

o Number of cells imaged per experiment

o How are foci defined in the analysis

o Exactly which cells are used in each experiment and were these cells damaged? (For instance, these details are lacking for figure 1D)

o Line 82: "We then used H2AX KO cells with a complementation system....." Please describe which complementation system you used specifically.

o Fig 3G: Where are the RNF168 KO cells coming from? If they are home-made, please include a western blot to show absence of RNF168. Otherwise, please add a reference.

Minor points:

o Fig 2B: the left panel is missing the labels.

o Line 198: "show binding affinity to the gH2AX C-terminal tail (Fig 3f)." Did the authors perhaps mean figure 5f instead of 3f?

o Line 338: "....when Y142 get re-phosphorylated and..." Do the authors mean de-phosphorylated instead of re-?

Reviewer #3 (Remarks to the Author):

In this manuscript, Kelliher et. al. first characterized the minimal DDR activating motif in H2AX (XSQEY), which is required for both MDC1 and 53BP1 recruitment. Then the authors identified Y142 mutation in H2AX, a known mutation that abolished the MDC1 recognition, still retain substantial ability to recruit 53BP1. Using biochemical approaches, the authors demonstrated that 53BP1 directly interact with H2AX, which requires a GKK—Q motif in H2AX and the oligomerization and tudor domain in 53BP1. Lastly, the authors indicate that H2AX C-terminal linker-region mediated 53BP1 recruitment is cell cycle regulated. Taken together, the authors presented a new model of 53BP1 recruitment to DNA damage sites, which fills a gap of current knowledge about the regulation of 53BP1 in DNA damage repair. Their data and methodology are mostly robust, and the quality of the work is generally acceptable. The novelty of this mechanistic finding is pretty high. However, I have several concerns, as discussed below:

1, Although the finding that H2AX can directly recruit 53BP1 independent of MDC1 is new and interesting, the functional importance of this regulation in DSB repair is not clear. Can the authors apply a reporter assay to show how much of the DNA repair efficiency (HR, NHEJ, or MMEJ) are recovered after reconstituting the Y142L mutant in H2AX KO cells.

2, In Fig. 4c-g, the authors show a GKK—Q motif in front of SQEY is essential for 53BP1 recruitment. In Fig. 1e, Lacking 120-137 (containing GKK—Q motif) still show strong MDC1 and 53BP1 recruitment. One can deduce the 53BP1 recruitment in Fig. 1e is solely dependent on MDC1, as GKK—Q motif is missing. Therefore, as add back of delta 120-137 can completely rescue the 53BP1 foci formation (Fig. 1f), MDC1 still seems to be predominant for 53BP1 recruitment. To further show the function importance of the MDC1 independent model in this manuscript, the authors can count the 53BP1 foci formation per cell in Fig. 1e, Fig 4c-g, as they did in Fig. 6a.

3. Fig. 3g is not clear. This data is important as it further establishes how important of the proposed

MDC1 independent model. Therefore, KO cells with overexpression of empty vector should be shown as a control. 53BP1 foci formation should be quantified, both cell numbers with >10 foci and foci numbers per cell.

4, In Fig. 6, the authors propose H2AX C linker region dependent 53BP1 recruitment is at G1 phase. This again need to characterize the function importance in DSBs repair. Like my point 1, can the authors apply the reporter assay to check the recovery of DNA repair efficiency in Y142L cells in G1 phase and S phase.

Response to reviewers

Reviewer #1 (Remarks to the Author):

gH2AX is the central player in DNA damage response and critical to sustain genome stability. Phosphorylation of gH2AX at Serine 139 has been widely accepted and used as a hallmark for damaged DNA. In this manuscript, Kelliher and colleagues present interesting results on the function of other surrounding gH2AX residues through the evolutionary perspective. After defining with the minimal DDR motif SQEY, Kelliher and colleagues found the Y142 is evolutionally divergent. Functionally, they found the Y142L is capable to recruit 53BP1 at the damaged chromatin independent from the MDC1 pathway. While the overall finding is new and might shed new light on how DNA repair signaling is activated across species, this reviewer feels that several important questions/experiments should be addressed to strengthen the conclusion before publication.

1 The author clearly demonstrated that overexpression of Y142L can induce 53BP1 foci. However, the downstream effectors like RIF1 after 53BP1 foci formation is not determined and this should be done to strengthen the conclusion.

Thank you for the great suggestion. We have added the RIF1 foci to strengthen the functional relevance of 53BP1 foci. Data were included in Supplementary figure 2 d-e.

2 In Figure 1b, the authors show F142 in Arabidopsis. Plants have different response to UV exposure and this F142 might explain why plants can survive the continuous UV exposure. This reviewer feel it is necessary to examine the 138F function in the same setting.

We did not include the Y142F mutant in our original study because Y142F has previously been shown to have a comparable affinity to MDC1 (Stucki M, Mol cell, 2005), and we believe it will be able to recruit MDC1. In the revised manuscript, we have reconstituted the Y142F mutant to H2AX KO cells and observed robust MDC1 foci formation (Supplementary Figure 2b). Therefore, we think the MDC1-RNF8-RNF168 signaling pathway is still active in the 142F mutant.

3 Does Y142L activate the BRCA1 pathway? A quick measure of BRCA1 foci formation should be done.

We have now included BRCA1 foci in Y142L reconstituted cells and we did not see any BRCA1 foci restoration. Data were included in Supplementary Figure 2i-j.

4 The 53BP1 formation after Y142L expression indicates the activation of NHEJ pathway. This can be determined by using the reporter assay after transient expression of Y142L mutant.

We have performed and included the EJ5-GFP reporter assay and found that Y142L transient overexpression showed a modest induction of NHEJ. Data were included in Supplementary Figure 5c.

5 The evolution of SQEY to activate DDR suggests this motif might be important during cancer development. Is any mutation identified within the SQEY region from the TCGA database? This might provide additional information about how SQEY evolution can reshape the DNA damage signal in cancer.

There is no mutation found within the SQEY region from the TCGA database.

6 Does Y142L overexpression change the drug sensitivity? Is any genome abnormality detected in Y142L overexpressed cells?

We have performed Cell-titer-Glo assay to determine if Y142L overexpression will change the CPT sensitivity, which is slightly different from our reconstitution system dataset. We believe that the Y142L mutant does not function as a dominant negative when endogenous H2AX is present.

7 The data from polyclonal antibody in figure 1d is a little confusing and is better removed.

Although we think the scientific community should know about the non-specificity of the polyclonal antibody that has been widely used, we agree with the reviewer's comment and removed the blot from the figure.

8 The title "Evolved histone tail regulates 53BP1 damaged chromatin recruitment" of the paper is confusing. This one might be more straightforward: "Evolved histone tail regulates 53BP1 recruitment at damaged chromatin".

We have edited the title per suggestion.

9 Since figure 2 has only 3 panels, it can be incorporated/combined with figure 3.

Thank you for the suggestion. We have merged a portion of figure 3, but not the full figure due to the space limit and combined some of figure 3 with figure 4 to improve the flow and clarity.

Reviewer #2 (Remarks to the Author):

Phosphorylation of H2AX is an important early step in the signaling of DNA damage, stimulating the recruitment of DNA damage repair proteins. MDC1 directly binds to and recognizes γ H2AX, which in turn leads to recruitment of the RNF8 and RNF168 ubiquitin E3 ligases, the subsequent ubiquitination of H2A and the binding of 53BP1 or BRCA1/BARD1, to coordinate repair. Kelliher et al. characterizes the C-terminal tail of H2AX and describes the minimal motif necessary to trigger the DNA damage response. In addition, the authors find recruitment of 53BP1 to ionizing radiation induced foci that is independent on MDC1 binding. These findings are of potential interest for the entire DDR field.

Thanks!

1. Kleiner et al (Ref.61) already demonstrated that 53BP1 can bind γ H2AX in the absence of MDC1. This reference should be cited much earlier in the manuscript and kept into account when drawing conclusions and providing novelty statements. The work in the submitted manuscript is still valuable in understanding this interaction at higher resolution, as they use different H2AX tail mutants. However, due to the comments below, there are concerns that needs to be addressed before publication.

Thank you very much for the comment. We have moved Kleiner et al reference up in the result part as our rationale why we tested the 53BP1 IRIF fragment but not the BRCT domain.

2. Functional assays to test the role of 53BP1 recruitment in the absence of MDC1 needs to be included.

We originally had the cell survival colony formation as our functional assay. In the revised manuscript, we have included comet assay (Supplementary Figure 6f) to demonstrate the function of DNA break clearance. We also included the DSB repair pathway choice assays in both overexpression and reconstitution settings (Supplementary figure 6c and e). Y142L showed a modest increase in distal end-joining efficiency but to a lesser extent comparing with the wildtype counterpart. Additionally, we have also included foci quantification of RIF1, a downstream 53BP1 protein, in Y142L reconstituted H2AX KO cells.

3. Does H2AK15Ub still occur (check by western blot)? Is it possible that this mark is placed independently on MDC1, explaining how 53BP1 is recruited?

Thank you for raising this important point. We have ectopically expressed RNF8 and RNF168 in H2AX KO cells. Due to the technical challenge of good antibodies, (in our experience, gammaH2AX antibody is the most sensitive to detect ub-and di-ub band), we were not able to detect the endogenous ubiquitination band. We then employed an approach that we have used in our previous publications, we used an SFB-tagged H2AX Y142L reconstituted H2AX KO cells with RNF8 and RNF168 overexpression and detect ubiquitination with the Flag antibody. In our western blot analysis, we were able to see an increase in ubiquitination in both RNF8 and RNF168 overexpression. Data were included in Figure 3c.

4. Are there other factors recruited at these sites?

Besides 53BP1 and its downstream effector protein RIF1, we do not have additional evidence to show any other factors are being recruited to these sites.

5. Does the linker-mediated 53BP1 binding in absence of MDC1 actually still stimulates NHEJ?

We have performed EJ-GFP reporter assay and we observed a modest increase in distal end joining efficiency. Data were included in Supplementary figure 6c.

6. What are the effects on longer timepoints?

We used 2Gy IR to quantify the 53BP1 foci . We observed that the foci were resolved at 24 and 48 hours. Data were included in Supplementary figure 6g

7. Any evidence of this interaction being important for DNA damage/repair pathways would really help. For example, are the mutant-complemented H2AX $-/-$ cells able to resolve damage?

Using comet assay, we observed that Y142L reconstituted cells could resolve DNA damage after 24 h similar to H2AX wildtype reconstituted H2AX KO cells. We also saw that Y142L complemented H2AX KO cells were able to resolve 53BP1 damage foci after 24-48 hours. Supplementary 6f-g

8. This is important for the interpretation of the data. For example: lines 318-323: “..... suggesting that γ H2AX linker-mediated 53BP1 recruitment requires and cooperates with a higher level of H4K20Me2 to achieve sufficient affinity....”. This seems too speculative without functional data.

We appreciate this comment. Although we are expressing our interpretation based on our data, we do agree that we are just speculating based on circumstantial evidence and previous reports (Pellegrino S. Cell reports, 2017), the actual data were not included in this study. We have rephrased the specific sentence to “We speculate that γ H2AX linker-mediated 53BP1 recruitment requires and cooperates with a higher level of H4K20me2” to improve clarity.

9. Concerns on the experimental setup and interpretations:

o The authors need to prove the presence of 53BP1 and lack of MDC1 on H2AX mutants, in the same IRIF co-localization experiment. Currently a lot of conclusions are drawn by separate experiments where only 1-2 proteins are stained (and there is not always a reference for foci formation, e.g. as in Figure 3B MDC1 staining). On this line, it is not clear how the authors define a foci in their image analysis. For example, in Fig 3D: MDC1 staining in H2AZ-SQEL looks like foci, and perhaps 53BP1 staining in macroH2A-SQEL as well. And in Fig 3B: Please include an image of a cell that doesn't have the 53BP1 foci or give us a bigger view with several cells, with and without foci. Including internal control or reference staining for foci helps to address these issues.

Thank you for the comments and we completely understand the concern. In our manuscript, we were trying to show the best-resolution immunofluorescence pictures. Due to the space limitation, we opted to present the best representative picture with quantification. In all of our experimental settings, we use H2AX KO + H2AX wildtype as a positive control and use either H2AX KO or H2AX KO + S139A as a negative control for MDC1 and 53BP1 foci. We also used image quantification software for foci quantification to verify our manual quantification and we obtained a similar result. Since the antibodies often have a background on staining which may be falsely counted as foci using software, we decided to quantify our foci manually.

Here, we attached several figures with a bigger view (of the original figure) as requested to demonstrate how we distinguish if the cells are forming foci. S. Fig. 1 showed that GFP-H2AZ SQEL and GFP-macroH2A-SQEL cells are not foci as the MDC1 and 53BP1 subcellular localizations are similar to their non-transfected H2AX KO cells. In contrast, the GFP-H2AX Y142L reconstituted H2AX KO cells displayed a visual difference between transfected and non-transfected H2AX KO cells. The pattern of the subcellular localization is similar to the GFP-H2AX wildtype reconstituted H2AX KO cells in S. Fig. 2. Which also has internal untransfected cells as negative control. We also included additional data for H2AX KO with GFP-H2AX Y142L reconstitution to show that Y142L, unlike H2AX wildtype, does not fully restore 53BP1 foci. Additionally, we have included several immunofluorescence co-staining micrographs and enlarged pictures for MDC1 and 53BP1 in both H2AX wildtype and Y142L stably expressed H2AX KO cells (S. Fig. 3) to show that 53BP1 IRIF formation can occur without MDC1 foci formation within the same cell.

I hope these can address the reviewer's concern.

Figure 3D

U2OS H2AX KO + GFP (10Gy 1h)

Figure 3B

A bigger view of the representative data presented in the original submission Figure 3 (original submission) and resubmission Figure 2

U2OS H2AX KO + GFP (10Gy 1h)

Additional data of immunofluorescence show that reconstitution of Y142L only partially restores 53BP1 foci and to show as an example how we discern cells with foci and no foci. We also use H2AX KO with GFP-wildtype reconstitution as a reference for foci quantification.

Additional immunofluorescence co-staining of 53BP1 and MDC1 in SFB-H2AX wildtype and SFB-H2AX Y142L reconstituted H2AX KO cells to show the presence of 53BP1 without MDC1 IRIF formation. We also included two additional enlarged immunofluorescence micrographs from independent experiments.

10. Are overexpression levels of H2AX mutants always the same? This needs to be quantified, as the authors show in figure 3G that overexpression of H2AX can by itself already induce the formation of foci. Therefore, the presence of foci can be dependent on the level of H2AX expression and not per se the mutation. It would be useful to bin foci formation based on GFP levels for the different H2AX mutants.

Thank you for this important point. Indeed, under the microscope, we observed a heterogenous level of expression of GFP on each reconstitution and we tried to average the foci quantification on each experiment. For negative control, such as the S139A mutant, almost every 53BP1 staining is negative in foci suggesting that this mutant is completely due to genetic regulation. On the other hand, in wildtype and other mutants, which showed intermediate phenotype (such as Y142L), there is a tendency of level-dependent, which is consistent with our claim in the discussion about levels of epigenetic marks (H4K20me2) does play a role in recruiting 53BP1 more efficiently.

We have included the binning quantification in Supplementary Figure 3.

11. 10Gy and 1 mM CPT are very high doses for inducing damage. Is the 53BP1 effect also present at lower levels of damage?

We have now included the 53BP1 foci quantification in Supplementary Figure 6a-b.

12. Many images have overexposure as well as strange histones or DNA staining. For example, figure 2B and 3D, both left panels have DAPI staining that is starkly different than expected. Figure 2B, middle panel, macroH2A is enriched in the nucleolus, which indicates it isn't in chromatin properly and therefore shouldn't be used to draw any conclusions from. Figure 4G, GFP signal is overexposed.

Thank you for pointing this out. As we mentioned above, every reconstitution has a heterogenous expression of GFP, to avoid confusion, we have replaced images with lower exposure, yet still showed the same phenotype. For the Hoechst staining, we reviewed the images and the original slides and found that the Hoechst signals were potentially faded when we took the pictures. To confirm this, we re-performed the experiment and took the images within 2 days and we did not see the nucleolus staining again. In the revised manuscript, the images were replaced.

13. Figure 4E: A control experiment with the WT H2AX sequences of these organisms is needed to show that WT fully rescues the effect. It is not possible to conclude with these data if the effect is due to the YtoL mutation or due to species differences.

We have now included the 53BP1 foci quantification for ortholog wildtype reconstitution in H2AX KO cells in Supplementary Figure 3g.

14. Figure 5D-H: The authors need to include a control with the gH2AX Y142L (and/or other) mutant peptide, to confirm the dependency of 53BP1 binding on the motif the authors have identified.

Mutant Y142L peptide pulldown experiment was included as suggested. Similar binding affinity to wildtype peptide was observed. Data were added in Figure 5g.

15. The authors include a part about the evolution of the H2AX tail, by performing complementation experiments in yeast. This part is hard to follow and confusing. The effects of

the mutants are really small and the conclusions of the authors on the presence or absence of a difference do not clearly match the colony growth results.

We agree with the comment. We have rewritten the result part to improve the clarity. Overall, we believe that the human histone does not fully function as yeast, even if we alter the C-terminus to mimic the yeast C-terminal tail. It may be due to additional genetic factors that we have not discovered.

16. Concerns with data presentation:

o It looks like data is reused for several figures. The WT and the Y142L look exactly the same in each graph (for i.e. % cells >5 53BP1 foci).

Thanks for pointing this out. We conducted some of the experiments/mutants at the same time with the wildtype control but presented them in different graphs for a better presentation purpose. We acknowledge that it may cause confusion. Here, we have performed additional wildtype controls in a separate setting and added them back to different graphs. Since we could consistently reproduce the same result, we hope it will address the reviewer's concern.

17. Supp fig 2D: This only shows two cells that were transfected, please add a quantification of more cells. Same for Supp Fig 3C.

Quantifications were added as recommended. In the new supplementary figure 2f and h, supplementary figure 4 d-e.

18. Line 139-141: " We ectopically expressed RNF8 or RNF168 in H2AX KO cells to elevate the ubiquitin signals on chromatin". Please demonstrate that there are indeed increased ubiquitin signals on chromatin in these cells.

Great point, see also reviewer 1's comment.

We have ectopically expressed RNF8 and RNF168 in H2AX KO cells. Due to the technical challenge of good antibodies, (in our experience, gammaH2AX antibody is the most sensitive to detect ub-and di-ub band), we were not able to detect the endogenous ubiquitination band. We then employed an approach that we have used in our previous publications, we used an SFB-tagged H2AX Y142L reconstituted H2AX KO cells with RNF8 and RNF168 overexpression and detect ubiquitination with the Flag antibody. In our western blot analysis, we were able to see an increase in ubiquitination in both RNF8 and RNF168 overexpression. Data were included in Figure 3c.

19. Line 155: " yHTA1 L132Y rescues both MDC1 and 53BP1 IRIF formation.....(Figure 4a,b)...." There is no data showing the rescue of the MDC1 in these conditions. Please include it or edit the statement.

We have edited the statement as suggested.

The method section needs to be expanded with the following details:
20 Antibody dilutions for IF and incubation time

We have added the dilution and incubation time in the method section.

21 Number of cells imaged per experiment

We have added the minimal number of cells imaged in the method section.

22 How are foci defined in the analysis

Foci were defined by comparing positive controls (wildtype or wildtype reconstituted KO cells) negative controls (KO/ untransfected cells or S139A reconstituted KO cells). The damage foci are visually discernible.

23 Exactly which cells are used in each experiment and were these cells damaged? (For instance, these details are lacking for figure 1D)

We apologize for missing the important information. We added the cell line back to figure 1D.

24 Line 82: "We then used H2AX KO cells with a complementation system....." Please describe which complementation system you used specifically.

We have added the description in the text. "In brief, we re-express H2AX (wildtype or mutant) with emGFP-expression vectors in U2OS H2AX KO cells. Since both MDC1 and 53BP1 do not form foci in H2AX KO cells³², we can use the untransfected cells as negative control while wildtype reconstitution serves as a positive control. "

25 Fig 3G: Where are the RNF168 KO cells coming from? If they are home-made, please include a western blot to show absence of RNF168. Otherwise, please add a reference.

We have added a reference in the text. The cell line is generated from our previous study.

Minor points:

1 Fig 2B: the left panel is missing the labels.

We added the labels back onto the figure.

2 Line 198: "show binding affinity to the gH2AX C-terminal tail (Fig 3f)." Did the authors perhaps mean figure 5f instead of 3f?

Sorry for the typo. We have fixed it accordingly

3 Line 338: "....when Y142 get re-phosphorylated and..." Do the authors mean de-phosphorylated instead of re-?

We have clarified this in the text. Y142L is constitutively phosphorylated, which suppresses MDC1 binding. It is dephosphorylated briefly after DNA damage to facilitate MDC1 binding

"The Y142 is constitutively phosphorylated by WSTF and transiently de-phosphorylated by EYA1 upon DNA damage, which facilitate MDC1 binding with γ H2AX. During the repair process, H2AX Y142 is being re-phosphorylated which limits MDC1 binding."

Reviewer #3 (Remarks to the Author):

In this manuscript, Kelliher et. al. first characterized the minimal DDR activating motif in H2AX (XSQEY), which is required for both MDC1 and 53BP1 recruitment. Then the authors identified Y142 mutation in H2AX, a known mutation that abolished the MDC1 recognition, still retain substantial ability to recruit 53BP1. Using biochemical approaches, the authors demonstrated that 53BP1 directly interact with H2AX, which requires a GKK—Q motif in H2AX and the oligomerization and tudor domain in 53BP1. Lastly, the authors indicate that H2AX C-terminal linker-region mediated 53BP1 recruitment is cell cycle regulated. Taken together, the authors presented a new model of 53BP1 recruitment to DNA damage sites, which fills a gap of current knowledge about the regulation of 53BP1 in DNA damage repair. Their data and methodology are mostly robust, and the quality of the work is generally acceptable. The novelty of this mechanistic finding is pretty high. However, I have several concerns, as discussed below:

1, Although the finding that H2AX can directly recruit 53BP1 independent of MDC1 is new and interesting, the functional importance of this regulation in DSB repair is not clear. Can the authors apply a reporter assay to show how much of the DNA repair efficiency (HR, NHEJ, or MMEJ) are recovered after reconstituting the Y142L mutant in H2AX KO cells.

Thank you for the suggestions, we have now included the repair assay data in supplementary figure 6e

2, In Fig. 4c-g, the authors show a GKK—Q motif in front of SQEY is essential for 53BP1 recruitment. In Fig. 1e, Lacking 120-137 (containing GKK—Q motif) still show strong MDC1 and 53BP1 recruitment. One can deduce the 53BP1 recruitment in Fig. 1e is solely dependent on MDC1, as GKK—Q motif is missing. Therefore, as add back of delta 120-137 can completely rescue the 53BP1 foci formation (Fig. 1f), MDC1 still seems to be predominant for 53BP1 recruitment. To further show the function importance of the MDC1 independent model in this manuscript, the authors can count the 53BP1 foci formation per cell in Fig. 1e, Fig 4c-g, as they did in Fig. 6a.

Great point, we totally agree with the reviewer, we do believe the MDC1-mediated pathway is predominant for 53BP1 foci. We have now included all the foci counting in the figures. Data were included in Supplementary figure 1c, Figure 4 d and h.

3. Fig. 3g is not clear. This data is important as it further establishes how important of the proposed MDC1 independent model. Therefore, KO cells with overexpression of empty vector should be shown as a control. 53BP1 foci formation should be quantified, both cell numbers with >10 foci and foci numbers per cell.

We have included the empty vector negative control and data were included in supplementary figure b. We have used >5 foci instead of >10 for consistency.

4, In Fig. 6, the authors propose H2AX C linker region dependent 53BP1 recruitment is at G1 phase. This again need to characterize the function importance in DSBs repair. Like my point 1, can the authors apply the reporter assay to check the recovery of DNA repair efficiency in Y142L cells in G1 phase and S phase.

We have included the reporter assay data. We do think that Y142L can modestly increase distal end-joining but the HR efficiency seems undistinguishable.

REVIEWERS' COMMENTS

Reviewer #1 (Remarks to the Author):

The authors have adequately addressed my questions and I recommend this article for publication in Nature Communication.

Reviewer #2 (Remarks to the Author):

The authors have sufficiently addressed the comments of the reviewers, we support publication of the manuscript.

Reviewer #3 (Remarks to the Author):

The authors have successfully addressed my concerns. I recommend this manuscript to be published in Nature Communications

Response to reviewers

REVIEWERS' COMMENTS

Reviewer #1 (Remarks to the Author):

The authors have adequately addressed my questions and I recommend this article for publication in Nature Communication.

Reviewer #2 (Remarks to the Author):

The authors have sufficiently addressed the comments of the reviewers, we support publication of the manuscript.

Reviewer #3 (Remarks to the Author):

The authors have successfully addressed my concerns. I recommend this manuscript to be published in Nature Communications

We would like to thank the reviewers for the constructive comments and suggestions which has significantly improved the our study.